# Currently Used Laboratory Methodologies for Assays Detecting PD-1, PD-L1, PD-L2 and Soluble PD-L1 in Patients with Metastatic Breast Cancer

**DOI:** 10.3390/cancers13205225

**Published:** 2021-10-18

**Authors:** Seri Jeong, Nuri Lee, Min-Jeong Park, Kibum Jeon, Wonkeun Song

**Affiliations:** 1Department of Laboratory Medicine, Kangnam Sacred Heart Hospital, Hallym University College of Medicine, Seoul 07440, Korea; hehebox73@hallym.or.kr (S.J.); nurilee822@hallym.or.kr (N.L.); mjpark@hallym.or.kr (M.-J.P.); 2Department of Laboratory Medicine, Hangang Sacred Heart Hospital, Hallym University College of Medicine, Seoul 07440, Korea; pourmythe45@hallym.or.kr

**Keywords:** breast cancer, checkpoint inhibitor, immunotherapy, metastasis, PD-1, PD-L1, PD-L2, programmed cell death, soluble PD-L1

## Abstract

**Simple Summary:**

Several methods targeting the programmed death protein-1 (PD-1) axis have been developed and evaluated for the detection of immune checkpoint levels that are strongly involved in immunotherapy for patients with metastatic breast cancer. Variations in different assays used in diverse studies have affected their result interpretation and clinical utility. When applying these assays to the laboratory, a comprehensive understanding of the characteristics of them should be recognized. We reviewed applied laboratory techniques for detecting PD-1, PD-ligand (L)1, PD-L2, and soluble PD-L1, which are important for selecting metastatic cancer patients for immunotherapy. Advances in methodologies according to the epoch are also investigated to gain insight into immunologic techniques and to facilitate appropriate laboratory settings for evaluating the PD-1 axis status, which are useful for estimating outcomes and planning patient-tailored immunotherapy strategies.

**Abstract:**

Approximately 20% of breast cancer (BC) patients suffer from distant metastasis. The incidence and prevalence rates of metastatic BC have increased annually. Immune checkpoint inhibitors are an emerging area of treatment, especially for metastatic patients with poor outcomes. Several antibody drugs have been developed and approved for companion testing of the programmed death protine-1 (PD-1) axis. We reviewed currently used laboratory methodologies for assays determining PD-1 axis to provide a comprehensive understanding of principles, advantages, and drawbacks involved in their implementation. The most commonly used method is immunohistochemistry (92.9%) for PD-L1 expression using tissue samples (96.4%). The commonly used anti-PD-L1 antibody clone were commercially available 22C3 (30.8%), SP142 (19.2%), SP263 (15.4%), and E1L3N (11.5%). Enzyme-linked immunosorbent assay and electrochemiluminescent immunoassay that target soluble PD-ligand (L)1 were developed and popularized in 2019–2021, in contrast to 2016–2018. Easy accessibility and non-invasiveness due to the use of blood samples, quantitative outputs, and relatively rapid turnaround times make them more preferable. Regarding scoring methods, a combination of tumor and immune cells (45.5% in 2016–2018 to 57.1% in 2019–2021) rather than each cell alone became more popular. Information about antibody clones, platforms, scoring methods, and related companion drugs is recommended for reporting PD-L1 expression.

## 1. Introduction

### 1.1. Epidemiology of Metastatic Breast Cancer

Among women, breast cancer (BC) is the most commonly diagnosed cancer and the second leading cause of cancer-related deaths in major countries [1,2]. According to the Global Cancer Observatory, 2,261,419 new breast cancer cases and 684,996 cancer-related deaths were estimated to occur worldwide in 2020 [3]. The BC incidence rates surpassed other cancers in both developed and developing countries [4,5]. The rates were highest in regions such as Australia, New Zealand, western and northern Europe, and North and South America [5]. In transitioning countries, the BC rates in South America [6], Africa [7], and high-income Asian countries [8] were found to rise rapidly. Women with BC living in transitioning countries have 17% higher mortality rates than those in transitioned nations [5]. This variability is related to sociocultural and/or dramatic environmental changes, including nutrition, hormone intake, lactation, and reproductive patterns [9]. Improved awareness of BC and clinical examination has led to its increased detection through mammographic screening. Due to the development of tumor markers and effective treatment, mortality rates in transitioned countries have decreased [10,11]. However, approximately 20% of BC patients suffer from distant metastasis within the first 5 years [12]. The incidence rate of metastatic BC in the United States increased by 2.5% annually from 2001 to 2011 according to population-based cancer data [13]. The number of women living with metastatic BC increased by 4% from 1990–2000, 17% from 2000–2010, and is projected to increase by 31% from 2010–2020 according to Surveillance, Epidemiology, and End Results (SEER) registries [14]. Patients with metastatic BC have a poor outcome, with an overall 5-year survival rate of 27% [13].

### 1.2. Immunotherapy and Immune Checkpoint of Metastatic BC

Most patients with metastatic BC received palliative and non-curative treatment. About 56% of patients were treated with radiation and chemotherapy alone, and 26% of them received no treatment or partial hormonal therapy [13]. However, targeted systemic treatments for BC, particularly for estrogen receptor (ER), progesterone receptor (PR), and human epidermal growth factor receptor 2 (HER2)-positive status [15], have improved survival for the metastatic disease over the past three decades [16]. Based on the SEER database, the 5-year relative survival rate had a 2-fold increase from 18% to 36% for patients diagnosed with de novo metastatic BC between 1992–1994 and 2005–2012, respectively. In the Netherlands, the 10-year overall survival rate increased from 6% to 9% for metastatic BC [17]. A previous meta-analysis also demonstrated that median survival increased for recurrent and de novo metastatic BC between 1990–2010 [16]. The survival rate of women with metastatic BC differed according to the ER status. For ER-positive patients, median survival increased from 32 to 57 months, and for ER-negative women from 14 to 33 months, reflecting the effectiveness of hormonal therapy. However, BC with negativity for ER, PR, and lack of amplification/overexpression of HER2, occupying about 10–20%, remained as an aggressive histologic subtype with worse outcomes [18,19]. Chemotherapies such as anthracyclines, taxanes, capecitabine, eribulin, and carboplatin are frequently used and recommended by the National Comprehensive Cancer Network (NCCN) [20] and the European School of Oncology-European Society for Medical Oncology (ESO-ESMO) [21] guidelines for patients with triple-negative BC. However, chemotherapies are generally related to unfavorable adverse events, and more so when combined, resulting in treatment discontinuation. Moreover, combining treatments did not prolong overall survival rates compared with monotherapies [22]. Therefore, the NCCN and ESO-ESMO guidelines [20,21] recommend the sequential use of single-agent chemotherapy for the treatment of metastatic BC, including the triple-negative subtype. In the recently published version (NCCN Version 3.2020 and 5th ESO-ESMO) of these guidelines [20,21], immunotherapy drugs are included as first-line therapy for programmed death ligand-1 (PD-L1)-positive metastatic BC. Immune checkpoint inhibitors are an emerging area of BC therapy, especially for patients with triple negative subtype. The PD-L1 checkpoint inhibitor atezolizumab, in combination with paclitaxel, has been related to prolong progression-free survival in a phase III trial (Impassion 130) [23] and has recently been granted approval by the US Food and Drug Administration (FDA). Furthermore, several novel monoclonal antibodies including pembrolizumab and nivolumab targeting programmed cell death protein-1 (PD-1) and durvalumab and avelumab targeting PD-L1 have been approved [24,25]. Furthermore, clinical trials evaluating combinations of immunotherapies and targeted therapies for all subtypes of BC have been conducted [26].

These recent investigations highlight the importance of testing immune checkpoint levels to select patients expecting greater benefit from these immunotherapies. Moreover, the drug approval process requires determination of PD-L1 expression for a combination of immune checkpoint inhibitors [27,28]. Upregulation of inhibitory immune checkpoint pathways by tumor cells (TC) and/or immune cells (IC) within the tumor microenvironment and the PD-1 axis enables TC to escape the immune system [29]. When PD-1 expressed on IC is activated by its ligands, PD-L1 or PD-L2, it attenuates lymphocyte activation and promotes the development of functional regulatory T cells, leading to the inhibition of the immune response [29,30]. As many tumors use this strategy to escape the immune system, the PD-1 signaling pathway has become an important therapeutic target in the field of breast cancer [31].

### 1.3. Aims of This Review

Several methods for the PD-1 axis have been developed and evaluated for the detection of immune checkpoint levels that are strongly involved in immunotherapy of patients with metastatic BC [25,31]. Variations in the assays used in diverse studies have affected their results and clinical utility. When applying these assays to the laboratory, a comprehensive understanding of the characteristics of them is required. Studies focusing on laboratory techniques for evaluating PD-L1 and related markers, including PD-1, PD-L2, and soluble PD-L1, have been rarely published. Therefore, our review describes several immunological assays for these immune checkpoints, the main determinants of immunotherapy in metastatic BC. Advances in methodologies according to the epoch have also been investigated based on published articles to provide an insight into the techniques and facilitate appropriate laboratory settings for evaluating PD-1 axis status, which is useful for estimating outcomes and planning patient-tailored immunotherapy strategies.

## 2. Sample Type

### 2.1. Tissue Sample

The sample types obtained from women with metastatic BC for testing PD-1 axis status are presented in Table 1. Twenty-eight of the studies were from three recently published systematic reviews and meta-analyses [22,32,33] and manual searching for the PD-1 axis in metastatic BC. The additional search strategy terms for PD-L2 and soluble PD-1 axis are as follows: (((Breast cancer[MeSH Terms]) OR ((breast) AND (cancer* OR tumor* OR tumour* OR carcinoma* OR neoplasm* OR carcinogen* OR malignan*))) AND (metastat* OR advance* OR second* OR recurren* OR inoperab* OR disseminat* OR incur*)) AND (pd-l2[Title/Abstract])) and (((Breast cancer[MeSH Terms]) OR ((breast) AND (cancer* OR tumor* OR tumour* OR carcinoma* OR neoplasm* OR carcinogen* OR malignan*))) AND (metastat* OR advance* OR second* OR recurren* OR inoperab* OR disseminat* OR incur*)) AND ((soluble pd-1[Title/Abstract]) OR soluble pd-l1[Title/Abstract]) OR soluble pd-l2[Title/Abstract])). Relevant articles with specific immunoassays for the PD-1 axis were manually checked. The most commonly utilized sample type was tissues (96.4%), including formalin-fixed paraffin-embedded (FFPE) tissues and freshly biopsied or surgically resected tissues (Figure 1). Archival FFPE tissues are a powerful resource for testing the PD-1 axis [34,35]. In particular, initial pre-metastatic tissue from metastatic BC patients for investigation was mostly FFPE because of its preservability [32]. However, perfect staining results cannot be guaranteed because of the possibility of tissue degradation and antigen loss in archived tissues over time. Therefore, a clinical trial included only biopsied or surgical specimens collected within 90 days [36]. Biopsies or surgically excised specimens should be fixed in neutral-buffered formalin for 6–72 h, depending on the sample size [37]. It is recommended that whole-slide sections with 2.5–4 μm thickness are cut from representative areas of the FFPE tumor tissue. The thickness depends on laboratory settings, such as targeting materials and commercially available autostainers. In general, the tissue sections were de-paraffinized with xylene and underwent rehydration and antigen retrieval for PD-1, PD-L1, and PD-L2 [34,35]. Tonsil tissue is recommended as a cost-effective positive control because the tonsil parenchyma contains PD-L1-positive immune cells.

### 2.2. Blood Sample

Serum (7.1%) and plasma (3.6%) samples were used simultaneously with the tissue samples. Blood samples were obtained according to commercially available kits used in the studies. For serum samples, approximately 3.5 mL blood was drawn for soluble PD-L1 measurements according to instructions from the Meso Scale Discovery platform by Domchek et al. [42]. For plasma samples, lithium heparin BD blood collection vials were used and promptly centrifuged within 4 h to obtain fresh plasma. The plasma specimens were stored at −80 °C until testing [47]. In general, the processing time and storage conditions of blood samples are critical for the stable and reliable measurement of target analytes [61]. Among the included studies, one study (3.6%) used blood samples only for circulating epithelial tumor cells [55]. Ethylenediaminetetraacetic acid (EDTA) tubes were used to collect 7.5 mL of blood samples. These samples were also processed promptly within 48 h of collection to obtain reliable results.

## 3. Targets

### 3.1. PD-1/PD-L1

The PD-1 pathway is a major checkpoint for immune responses in the tumor microenvironment [62]. PD-1 (CD279) is an inhibitory co-receptor presented on the surface of T lymphocytes [63]. PD-L1 (also known as B7-H1; CD274) is mostly expressed on infiltrating immune cells [64]. The binding of PD-L1 to PD-1 leads to the inhibition of the T cell response, enabling self-tolerance and immune tolerance to prevent excessive immune reactions [65]. In malignant condition, cancer cells also present PD-L1 to inactivate and exhaust the T cell activation [37,66]. This mechanism involves immune escape or local suppression of the immune system (Figure 2). In BC, the expression of PD-1 is related to an aggressive phenotype including a high tumor grade and lack of ER expression [67]. The rate of PD-L1 expression by TC in triple-negative BC ranged from 19% [68] to 59% [69]. Both PD-1 and PD-L1 positive cells are observed more often in triple-negative BC than in other subtypes, resulting in worse outcomes in patients [69,70]. Blocking the PD-1/PD-L1 interaction through therapeutic monoclonal antibodies, known as immune checkpoint inhibitors, reduces immune suppression and generates the activity of tumor-specific T lymphocytes. Drugs that antagonize the PD-1 axis have demonstrated clinically effectiveness in patients with advanced tumors [33]. Therapeutic monoclonal antibodies such as pembrolizumab and nivolumab for PD-1, and atezolizumab, durvalumab, and avelumab for PD-L1 have been trialed and approved by the FDA [71]. Therefore, PD-1/PD-L1 expression by TC or infiltrating IC is under investigation as a predictive biomarker for response to blockade of the PD-1 axis. Among the targets of PD-1 axis, all included studies identified PD-L1 (100.0%) (Figure 3A). Other targets simultaneously tested with PD-L1 were PD-1 (22.2%), PD-L2 (33.3%), and soluble PD-L1 (33.3%) (Figure 3B). One study investigated all PD-L1, PD-1, and PD-L2 expression concurrently.

### 3.2. PD-L2

The second ligand for PD-1 is PD-L2; however, its role in modifying immune responses is unclear and available information is limited. PD-L2 is usually presented at a lower level than PD-L1. However, the relative affinity of PD-L2 to PD-1 is 2 to 6-fold higher than that of PD-L1 because of direct interactions of PD-L2 with PD-1 [72]. Tryptophan (W110_PD-L2_) unique to PD-L2 contributes to different affinities. The W110_PD-L2_ plays the role of “elbow” and a C–D region functions as “latch”, potentiating the affinity advantage of PD-L2 structurally [73,74]. In general, PD-L2 is expressed on immune cells including macrophages and dendritic cells [75]. Signaling pathways downstream of cytokine receptors and innate immune activators regulate PD-L2 expression. PD-L2 mainly plays a role in the induction phase of Th2-driven T-cell immune reactions. Tumor microenvironments usually deviate towards an ineffective Th2 type of immune condition, leading to cancer cells escaping from immune surveillance. Therefore, studies on the relevance of PD-2 in metastatic BC have been conducted [52,55,72]. PD-L2 accounted for 44.4% of the other targets simultaneously measured with PD-L1 (Figure 3B).

### 3.3. Soluble PD-L1

Soluble PD-L1, which can attach to PD-1, is usually generated following proteolytic cleavage of membrane-bound PD-L1 by translation of alternatively spliced mRNA or matrix metalloproteinase [76]. Soluble PD-L1 has been reported to be a potential prognostic predictor in several cancers [47]. Soluble PD-L1 indicates an anti-immune response and cooperatively affects tumor progression. It is also associated with immune suppression through the regulatory function of T lymphocytes [77]. In addition to clinical significance, the expression of soluble costimulatory materials has been reported to be correlated with clinicopathological features such as lymph node and multiple organ metastasis. Therefore, soluble PD-L1 is a useful predictor in recurrent or metastatic BC before receiving the first line of therapy. High levels of soluble PD-L1 are related to shorter progression-free survival [78]. Regarding immunotherapy, therapeutic anti-PD-L1 might be absorbed by soluble PD-L1 before reaching TC, which alters the killing effect of BC cells on IC [60]. Meanwhile, a specific anti-PD-L1 antibody agent might suppress the expression of soluble PD-L1 and remove its blocking effect on the PD-1 negative signaling pathway, leading to improving T cell viability and enhancing the killing of BC cells [47]. There have been several studies investigating soluble PD-L1 because abnormalities in this factor are involved in early and long-term immune modulation. The soluble form of PD-L1 consists of 33.3% of the other targets assessed with PD-L1 (Figure 3B).

## 4. Anti-PD-L1 Antibody Clone

A total of nine types of anti-PD-L1 antibody clones were identified in studies on PD-L1 expression in metastatic BC (Figure 3C). Among clones occupying more than 10%, PD-L1 22C3 was used most commonly (30.8%), followed by SP142 (19.2%), SP263 (15.4%), and E1L3N (11.5%). The binding epitope of the 22C3 clone is a discontinuous segment of the extracellular domain [79]. The binding site of 22C3 contains N-linked glycosylation, which may influence the binding efficacy of antibodies and is related to variability in antigen retrieval. Meanwhile, SP142, SP263, and E1L3N bind to the C-terminal cytoplasmic domain [25]. Cytoplasmic staining may show a punctate or granular pattern [80]. Studies comparing PD-L1 assays in triple-negative BC demonstrated discrepancies among assays using 22C3, SP142, and SP263 clones [53,81]. The results of the 22C3, SP263, and E1L3N assays were broadly comparable, whereas SP142 showed lower PD-L1 expression on TC [53,81,82]. The assays with 22C3 and SP263, which have lower sensitivity for IC, mainly stained macrophages and dendritic cells.

Meanwhile, SP142 stained IC, such as some CD68 positive lymphocyte-like cells [83,84]. IC staining was easily identified using the SP142 assay because of the lower prevalence of TC staining. Therefore, SP142 showed higher inter-reader agreement for studies with PD-L1 IC > 1% [82,85]. The clinical trial (IMpassion 130) [23] evidencing the FDA approval of the VENTANA platform as a companion test for atezolizumab in triple-negative BC, adopted SP142 clone available only in VENTANA Medical Systems. The scoring method applied to this study was IC, which is unique to SP142 clone. Regulatory agencies have approved only the SP142 assay as a complementary diagnostic test for the administration of atezolizumab in countries such as the United States, Japan, Sweden, Peru, and Argentina. In particular, only the SP142 assay is covered by the health system of Japan because regulatory agencies have approved it as a companion diagnostic test for triple-negative BC. On the other hand, the European Union (EU), China, and Brazil permit validation of the PD-L1 assay. Drugs are not mandatorily correlated with companion assays in the EU. The lower prevalence of PD-L1 positivity determined by the SP142 clone could potentially reveal fewer patients selected for immunotherapy. Meanwhile, the use of 22C3 or SP263 could lead to greater patient eligibility. However, there are still concerns about drug toxicity and financial costs without clinical benefit because of false-positive results.

In the most commonly used 22C3 clones in metastatic BC, pembrolizumab for PD-L1 (22C3)-selected metastatic triple-negative BC patients might be approved based on the conclusions of an expert round-table discussion [31]. Laboratories performing PD-L1 testing for lung cancer have already used other assays, most frequently 22C3 and SP263 assays [86,87]. Some laboratories could have difficulty developing and validating the SP142 assay. Further, commercial diagnostic assays for 22C3 are available and can be performed on different platforms. In addition, PD-L1 testing could be performed on other clones, such as SP263 and E1L3N, if analytically validated and permitted by regulatory agencies. Based on our data, 22C3 assays were frequently performed on a platform from Agilent and Merck & Co. of the United States and Q2 solutions of the UK. Meanwhile, assays for SP142 and SP263 were performed on a platform from Ventana Medical Systems. For E1L3N, clones from cell signaling were frequently used to detect PD-L1 expression (Table 1).

## 5. Applied Methods

### 5.1. Immunohistochemistry (IHC)

IHC is the most commonly used technique for the detection of the PD-1 axis in patients with metastatic BC (Figure 4A). The antibodies selectively binding to target antigens such as PD-1, PD-L1, and PD-L2 is essential for proper protein identification from fresh and appropriately processed tissue [88]. Antibodies conjugated to an enzyme enable the visualization of an antibody–antigen complex [89]. Commercially available PD-L1 assays, adopting specific antibody clones, are commonly utilized in some studies for metastatic BC. Each assay using a specific score and cut-off has been applied to these studies targeting specific immune checkpoint inhibitors. Furthermore, commercial assays have been validated using specific platforms. Providing specific information about the adopted assay, antibody clones, related platforms, and validated in-house protocols used in clinical laboratories is necessary, especially if the applied immune checkpoint inhibitors are associated with companion diagnostic tests for the PD-1 axis. FDA approval of some immunotherapies is indeed associated with specific companion assays [23,71]. On the other hand, many laboratories in European countries have established laboratory-developed tests according to current practices in the European Union (Table 1). Therefore, the manufacturer’s recommendations, established in-house protocols considering the regional guidelines and accredited institutions are important for testing the IHC of PD-L1 [37]. For the most commonly used 22C3 clone, several IHC detection systems such as Link 48 autostainer and EnVision Flex from Agilent and OptiView DAB IHC detection kit from Ventana Medical System have been applied [25,37,58]. Meanwhile, the SP142 clone has been mostly associated with the OptiView DAB IHC detection kit from Ventana Medical System [40]. In terms of SP263, Ventana BenchMark Ultra autostainer using the OptiVew DAB IHC detection kit from Ventana Medical System was also commonly used [49,57]. For the E1L3N clone, Leica Bond-Max Autostainer (Leica Biosystems) was used for PD-L1 IHC [46]. The advantage of IHC is its ability to reveal the exact location of a targeting antigen within the tissue. Furthermore, the procedure requires only a small amount of tissue and is relatively simple and inexpensive. However, the results should be interpreted cautiously as this method is semi-quantitative and various scoring methods and cutoffs are applied to the results [90]. Harmonization studies comparing different staining and counting methods are needed, particularly for metastatic BC.

### 5.2. Enzyme-Linked Immunosorbent Assay (ELISA)

Studies that use plasma or serum samples for soluble PD-L1 detection have adopted ELISA (Figure 4B). In these immunoassays, enzymes are conjugated to secondary antibodies, which bind to the soluble PD-L1 antigen–antibody complex. The conjugated enzyme catalyzes the production of a colored end product after the addition of appropriate substrate and incubation. The catalyzed products can be visualized and quantified [91]. Distinguishing specific soluble PD-L1 antigens from nonspecific complexes is required for most commercial enzyme immunoassay systems such as ELISA. Regarding sandwich ELISA, which is commonly utilized to identify soluble PD-L1, plasma or serum specimens are firstly added to a well coated with an immobilized antigen-specific antibody. If the soluble PD-L1 is present, it will attach to the antibody. After washing out, a secondary antibody targeting the soluble PD-L1 is added. An enzyme conjugated to this secondary antibody directly or the third conjugated antibody to the Fc region of the second antibody can be catalyzed to produce the color reaction when the substrate is added. The amount of color is proportional to the amount of targeted soluble PD-L1 in the specimen. When used to identify soluble PD-L1, the targeted materials of soluble PD-L1 are “sandwiched” between two monoclonal antibodies. Absorbance is determined at a specific wavelength using a spectrophotometer. The concentration is calculated through a standard curve plotted from the known concentrations of the ligand. The kits from USCN Life Science and R&D system used for the determination of soluble PD-L1 in metastatic BC studies were sandwich ELISA assays [47,60]. Due to the easy accessibility and non-invasiveness of blood samples, ELISA-based tests have been applied to the measurement of soluble PD-L1. In addition, providing quantitative values and relatively rapid turnaround times with shorter hands-on time makes this technique available in several laboratories. Further studies are needed to validate these blood sample-based methods. Although IHC of tissue is widely used as a reference, these assays might be more popular because of the advantages of detecting soluble targets.

### 5.3. Electrochemiluminescent Immunoassay (ECL)

One study on soluble PD-L1 in metastatic BC used an ECL. This technique utilizes electrochemical compounds as labels to generate light from an oxidation-reduction reaction. Ru(bpy)_3_^2+^ regenerated with tripropylamine is used for ECL for superior performance. It can be a useful imaging tool because of the inherent luminosity [91,92]. The generated photons are identified using photomultiplier tubes, silicon photodiodes, or gold-coated fiber-optic sensors. Its high sensitivity, low background, and easily controlled characteristics make this technique preferable [93,94]. In addition, the variation in the electrode potential improved the selectivity of this method [95]. Regarding soluble PD-L1, the mesoscale discovery platform adopted the ECL technique employing biotinylated anti-PD-L1 capture antibody clone 2.7 A4 [42]. During incubation, soluble PD-L1 in the serum sample bound to anti-PD-L1-specific antibodies immobilized on a streptavidin-coated 96-well plate. After the unbound substances were removed, the primary antibody was identified by the addition of a ruthenium-labeled secondary antibody. Chemiluminescent emission after the application of voltage to the electrode was detected using a photomultiplier tube. The values were determined using a calibration curve. The subject-based limit of detection was 67.1 pg/mL and the quantitation range was 15.6–1000 pg/mL.

### 5.4. Fluorescence Immunoassay

Fluorescence immunoassays generally use fluorescently labeled antibodies [96]. When a fluorophore is excited, light at a specific wavelength is generated. A fluorometer provides the excitation light source and the immunofluorescence microscopy or photomultiplier tube detects the emission fluorescence. The selection of substrates without substances emitting fluorescent light is important for improving the sensitivity of this assay [97]. In addition, the correct absorption wavelength is necessary to excite the fluorophore tag attached to the antibody and to identify the released fluorescence. Schott et al. [55] adopted a fluorescence immunoassay for circulating epithelial tumor cells to detect PD-L1 and PD-L2. Cells with green, red, and blue surface staining and well-preserved nuclei were visualized. The results for PD-L1 and PD-L2 were calculated as a percentage of the total number of circulating tumor cells.

### 5.5. Quantitative Real-Time PCR

One study used quantitative real-time PCR to investigate the immunogenomic dynamics of metastatic BC [55]. The expression levels of immune-related genes in the biopsied samples were measured by quantitative real-time PCR. Another study on the methylation of PD-L1 was also performed using this technique [46]. Real-time PCR is commonly utilized for the analysis of gene expression and quantification [98]. Nucleic acid extraction and purification are the basic procedures for this technique. Bisulfite conversion of DNA was performed for methylation analysis. The target materials were amplified using PCR. The PCR thermal cycling program is composed of denaturation, annealing, and elongation. Quantification can be conducted using real-time PCR. The simultaneously performing reverse transcription PCR and real-time PCR enables combined nucleic acid amplification and detection in a single step [99]. Real-time PCR assays utilizing target-specific probes to detect amplified products is generally adopted in clinical laboratories. TaqMan probes are commonly used as fluorescent probes for real-time PCR. The probes hybridize to the target sequences and cleave the fluorescent probe during the PCR amplification step. The decoupling of the quencher material and fluorescent probe improved the fluorescence intensity [100]. The disadvantages of TaqMan probes are that they are time-consuming, expensive, and require separate probes for each RNA target. In addition, the spectral overlap of fluorophores should be cared about, especially for fluorescent PCR assays. Confirming the specificity of probe hybridization with caution when considering its relative convenience and usefulness. These TaqMan probes were used to detect the *PD-L1* and *PD-L2* genes in metastatic BC. Although quantitative real-time PCR is highly sensitive, RNA degradation, contamination, false-positive results due to non-specific amplification, and false-negative results due to lower levels of gene expression should be considered.

## 6. Scoring Methods

In the evaluation of PD-L1 expression to select patients for immunotherapy, various cell types and different cutoffs have been applied to studies of metastatic BC (Figure 5). The combination method, including both TC and IC, consisted of 52% of all included scoring methods. The following methods were used to count the TC (36%) and IC (12%). PD-L1 expression by IHC was evaluated semi-quantitatively by a pathologist. Because of the subjective aspects of interpretation and poor reproducibility, analysis by at least 8–10 pathologists was recommended for reproducibility of PD-L1 expression according to a summary of an expert round-table discussion [31]. Moreover, training in PD-L1 assessment for pathologists annual internal and external quality assurance is recommended [25,101]. In general, TC is counted as PD-L1 positive if membranous staining is present. If there is cytoplasmic staining without membranous staining, the TC is considered PD-L1-negative. In contrast, either granular cytoplasmic or incomplete membranous staining can be designated as a positive count for IC [66,80,85]. IC, including granulocytes, lymphocytes, dendritic cells, and macrophages, can be found in clusters or dispersed single cells. The presence and distribution of ICs in the tumor area should be recorded. At least 50–100 TC and associated stroma should be presented, and the whole tumor area without necrotic areas must be assessed. Staining artifacts, necrosis, or intravascular IC were excluded from the assessment. The scoring methods were varied and classified into TC, IC, and the combination of TC and IC. First, TC was composed of tumor cell and tumor proportion scores. The tumor cell score is defined as the percentage of the area composed of PD-L1 positive TCs in the entire tumor area [102]. Meanwhile, the tumor proportion score defines the ratio of PD-L1 positive TC, relative to all TCs, multiplied by 100% [103]. Second, for immune cell score, all IC located in the intra-TC region or peri-tumor stromal rim were considered when calculating the score. The percentage of the area occupied by all PD-L1 positive IC relative to the whole tumor area, including TC and associated, was counted. Lastly, the combined positive score involves both TC and IC located in the TC or the narrow rim around the TC. The number of PD-L1-positive TCs and ICs was counted, relative to the number of all vital TCs, and then multiplied by 100 [102,104]. These scoring methods were associated with the adopted clones. The SP142 assay usually counts the IC. Meanwhile, 22C3, SP263, and E1L3N determine TC and the combination of TC and IC, as well as IC only. Therefore, the proportions of combination and TC were higher than those of IC in the included studies (Figure 5). Regarding the cutoffs, 80% of the studies (20 out of 25 studies utilizing IHC with reported cutoff values) adopted a cutoff value of 1% (Table 1). In the TC scoring method, two studies applied a 5% cutoff [41,46]. For IC and combination scoring methods, a 10% cutoff was applied to the IC of three studies [36,52,57]. Because of this heterogeneity, reports on PD-L1 detection should contain information about the antibody clone, the applied scoring method with cutoff, and the relevant therapeutic setting of the respective tumor and approved drug. Additionally, the staining assay and platform used can also be described [25,102].

## 7. Trends in Applied Assays for Metastatic BC

The changes in samples for identifying the PD-1 axis in metastatic BC according to the period of examination are presented in Figure 6. The use of blood samples (20.0%) in 2019–2021 was higher than that in 2016–2018 (7.7%). Measurement of soluble type of PD-L1 with samples having easier accessibility and non-invasiveness could influence changes in user preferences. There were no changes in the targeting substances for PD-L1 between 2016–2018 and 2019–2021 because all included studies measured PD-L1 expression (Figure 7A). PD-L1 would be assayed essentially as a companion test when immune checkpoint inhibitors are prescribed. For targets other than PD-L1, PD-L2 was commonly determined in 2016–2018; it is no longer used in 2019–2021. PD-L2 is expressed in only a minority of patients compared to PD-L1 [55]. Furthermore, the role of PD-L2 in modulating immune responses is less obvious, which leads to a lack of targeted drugs [72] and a decreased number of tests in 2019–2021. PD-L2 would be mainly assayed for research use only rather than clinical practice. In contrast, PD-1 could be tested when immune checkpoint inhibitors targeting PD-1, such as pembrolizumab and nivolumab, are administered to BC patients. The advent of kits for soluble PD-L1 led to the differences in the targets between 2016–2018 and 2019–2021 (Figure 7B). The number of anti-PD-L1 antibody clones decreased from 7 to 5 between 2016–2018 and 2019–2021 (Figure 7C). Commonly used clones, such as 22C3 and SP142, were consistently used in 2019–2021 and 2016–2018. The proportions of 22C3 and SP142 in 2019–2021 increased because of the use of commercialized PD-L1 kits rather than in-house assays. In particular, the wider use of the SP142 clone is related to clinical trials such as IMpassion130 trial [105] and led to regulatory agencies mandating specific companion diagnostic tests. In addition, the use of SP263 associated with durvalumab and E1L3N was also reflected in the clones from 2019–2021. For the detection method, IHC became an essential assay in 2019–2021 (Figure 8A). There have been no studies without IHC in recent times. Additionally, for the changes in other detection methods, ELISA and ECL, which are capable of quantitative measurement of soluble PD-L1, were introduced and became the main techniques (Figure 8B). The proportion of combination scoring increased from 45.5% in 2016–2018 to 57.1% in 2019–2019 (Figure 9). These changes might have occurred because of the increased proportion of 22C3 clones concurrently counting IC and TC.

## 8. Future Perspectives

Detection of PD-L1 using IHC has been widely used for selecting the best candidates for immunotherapy [71,106]. The procedure needs to be effective and reproducible through further concordance studies, standardized protocols, and guidelines. In addition, studies are needed to identify biomarkers with strong predictive values. Soluble PD-L1 is beneficial for TC to resist the killing and elimination of lymphocytes in the tumor microenvironment. Although soluble PD-Ll has been recognized as a naturally occurring modulator of PD-1 signaling pathways, its biological activity remains poorly understood. Further studies are needed to better understand the implications of soluble PD-L1 levels on immunotherapy efficacy in metastatic BC. Moreover, diverse methods have been utilized for the enrichment and detection of circulating tumor cells [107,108]. The developed techniques demonstrated high sensitivity and facilitated the quantification of these cells [109,110]. However, analyses of images can be subjective, and therefore, data should be interpreted with caution because of the high background fluorescence related to this methodology. Sophisticated techniques for detecting the PD-1 axis are currently being developed for future use. The rapid and accurate detection based on the advent of newly developed techniques will substantially benefit patients receiving immunotherapy. Furthermore, the genetic profiling of the PD-1 axis using molecular tools such as microarrays and next-generation sequencing might increase the accuracy of predicting responses to immunotherapies. Analytical and clinical validation should be conducted before using new techniques in clinical laboratories. Furthermore, targeting only the PD-1 axis alone cannot explain the complete restoration of T cell function [111]. Therefore, the identification of other substances involved in T cell dysfunction is also required. Tumor infiltrating lymphocytes (TILs), a prognostic biomarker in various types of cancers, has been reported to be related to the prognosis of triple-negative BC [3,112,113]. Recently, CD73 expressed on cancer and immune cells has been investigated as a molecular immunosuppressive factor in triple-negative BC [112]. CD73 can increase extracellular adenosine by converting adenosine monophosphate to adenosine, which activates high affinity A2A and A2B adenosine receptors leading to immune suppression [113]. Therefore, reports have associated CD73 overexpression with anthracycline resistance [114] and worse prognosis in triple-negative BC [112,115]. Although PD-1 axis is one of the main mechanisms for immune escaping, the prognostic implications of PD-L1 are limited [116,117]. The combination of these other markers with PD-1 axis could be proposed as a useful predictive tool [115]. In addition, further studies investigating the effect of anti-CD73 antibody therapy on reprogramming the tumor microenvironment would contribute to better outcomes of metastatic BC.

## 9. Conclusions

In conclusion, we reviewed the applied laboratory techniques for detecting PD-1, PD-L1, PD-L2, and soluble PD-L1, which are important markers for selecting metastatic BC patients for targeted immunotherapy. In laboratories, the main method used for PD-1 axis is identifying expression in IHC with tissue samples. The most commonly used anti-PD-L1 antibody clones are 22C3, SP142, SP263, and E1L3N. Assays targeting soluble PD-L1, such as ELISA and ECL, have been developed and have been widely utilized. Easy accessibility and non-invasiveness due to the use of blood samples, providing quantitative values, and relatively rapid turnaround times with shorter hands-on time make these techniques preferable in several laboratories. For the scoring methods, the combination of TC and IC, rather than each cell alone, became more popular in correlation with the increased use of 22C3. To use these assays, it is important to have a comprehensive understanding of the associated principles, advantages, disadvantages, and precautions for accurate data interpretation. In particular, providing information about utilized antibody clones, platforms, scoring methods, and related companion drugs is recommended for reporting PD-L1 expression.

## Figures and Tables

**Figure 1 cancers-13-05225-f001:**
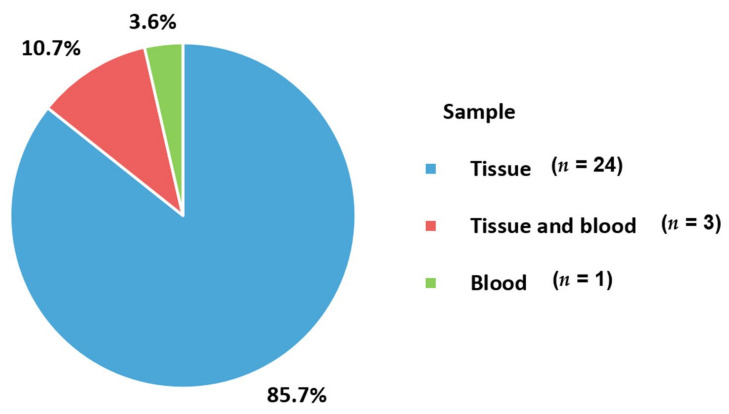
Sample type for identifying PD-1 axis.

**Figure 2 cancers-13-05225-f002:**
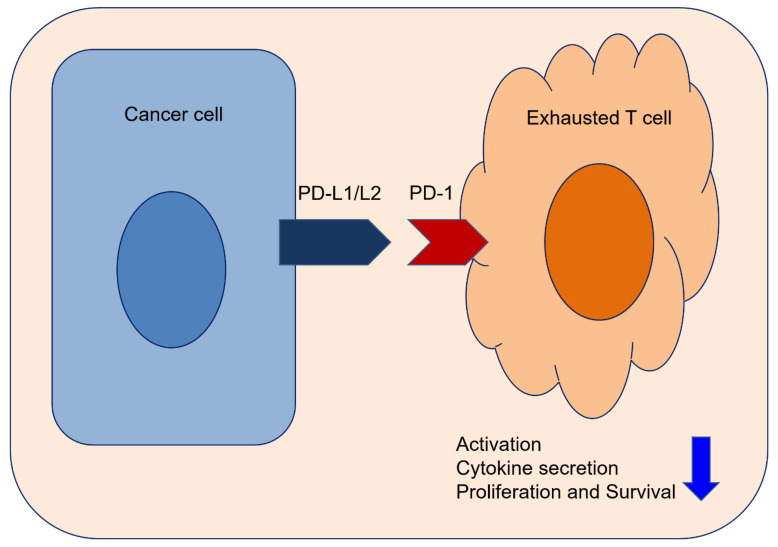
PD-1 axis exhausted T cell leading to immune escape.

**Figure 3 cancers-13-05225-f003:**
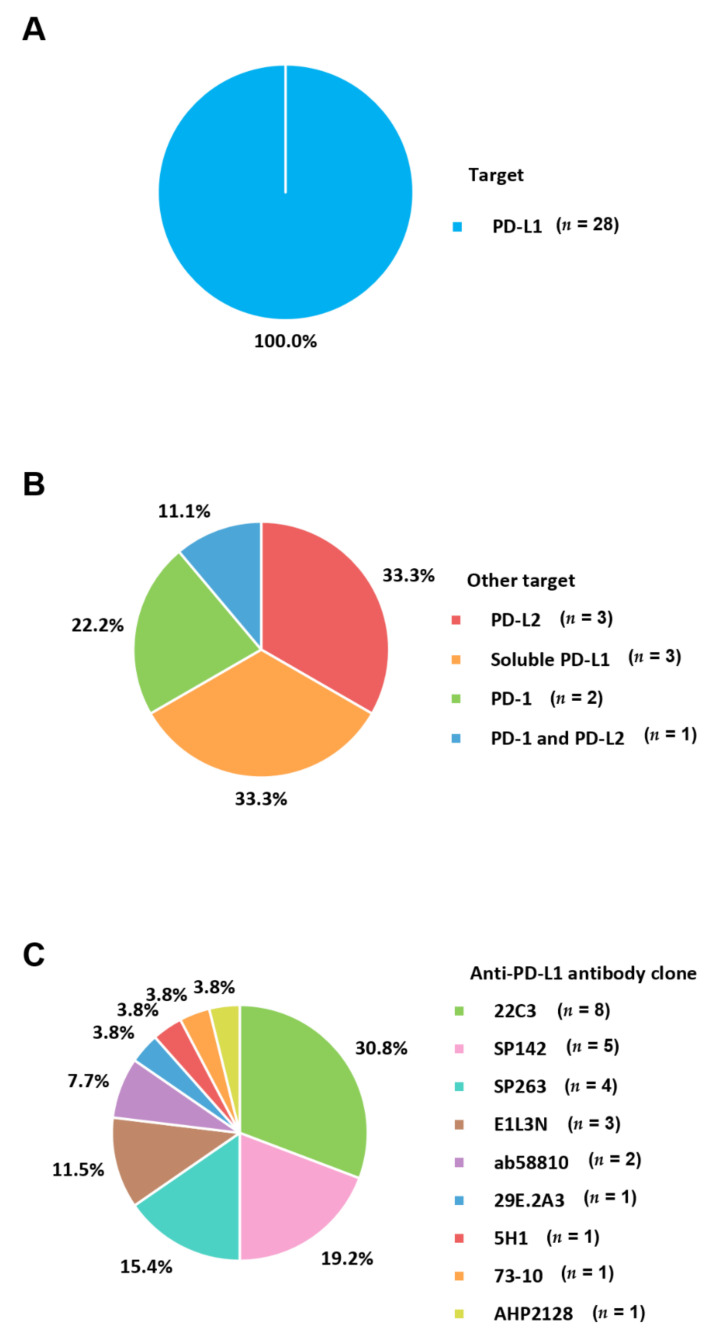
Targets of PD-1 axis and antibody clones for PD-L1. (**A**) Pie charts showing the target of PD-L1; (**B**) pie charts presenting other targets concurrently used with PD-L1; and (**C**) pie charts showing the antibody clones of PD-L1. PD-1, programmed cell death protein-1; PD-L1, programmed death ligand-1; PD-L2, programmed death ligand-2.

**Figure 4 cancers-13-05225-f004:**
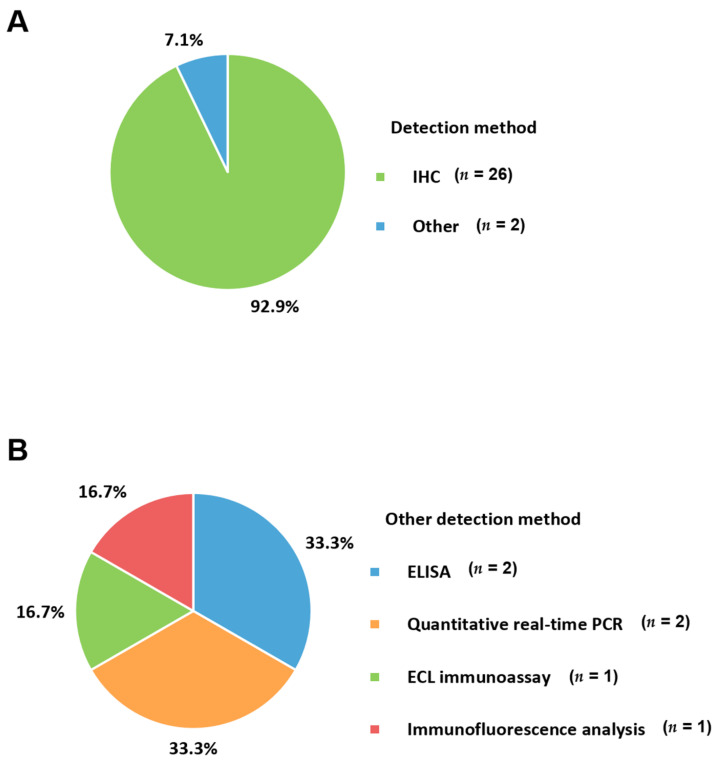
Applied techniques to detecting the PD-1 axis. (**A**) Pie chart representing the detection method; (**B**) pie charts showing other detection methods for the PD-1 axis. PD-1, programmed cell death protein-1; IHC, immunohistochemistry; ECL, electrochemiluminescent immunoassay; ELISA, enzyme-linked immunosorbent assay.

**Figure 5 cancers-13-05225-f005:**
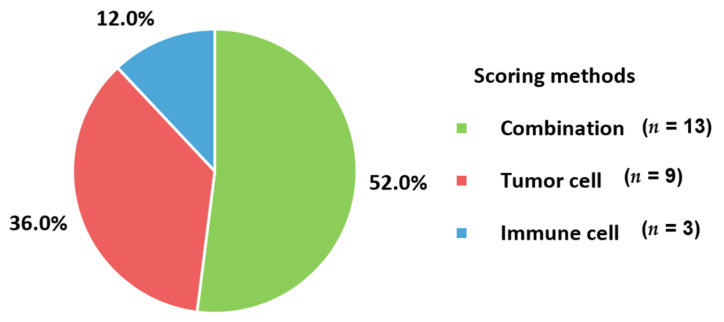
Applied scoring methods to identifying PD-1 axis. PD-1, programmed cell death protein-1.

**Figure 6 cancers-13-05225-f006:**
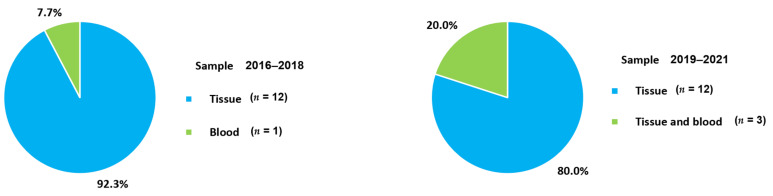
Changes in the used sample types according to the epoch.

**Figure 7 cancers-13-05225-f007:**
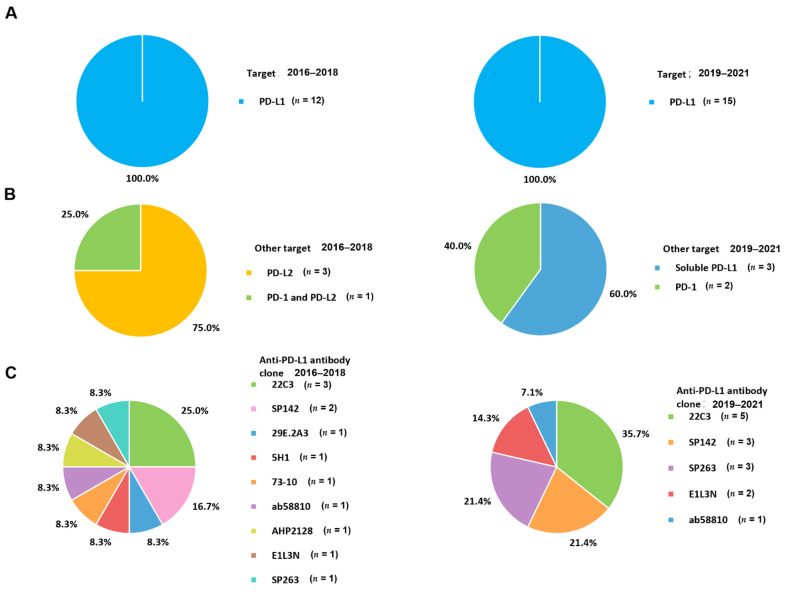
Changes in the targets for PD-1 according to the epoch. (**A**) Pie charts showing the consistent target of PD-L1; (**B**) pie charts presenting the changes of other targets concurrently used with PD-L1; and (**C**) pie charts showing the changes of antibody clones of PD-L1. PD-1, programmed cell death protein-1; PD-L1, programmed death ligand-1; PD-L2, programmed death ligand-2.

**Figure 8 cancers-13-05225-f008:**
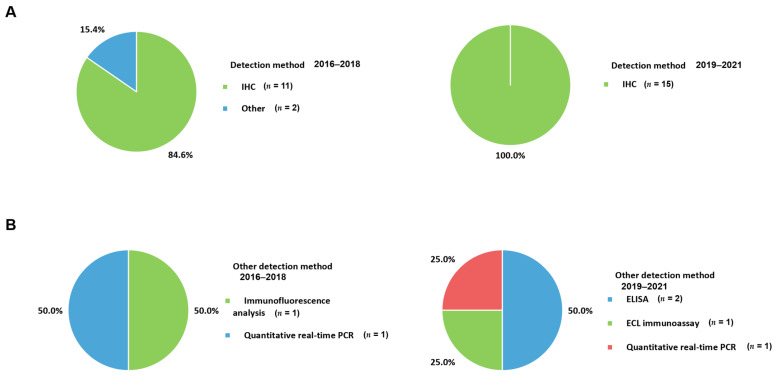
Changes in applied methods for detecting the PD-1 axis. (**A**) Pie charts presenting the changes in detection method; (**B**) pie charts showing the changes in other methods for detecting the PD-1 axis. PD-1, programmed cell death protein-1; IHC, immunohistochemistry; ECL, electrochemiluminescence; ELISA, enzyme-linked immunosorbent assay.

**Figure 9 cancers-13-05225-f009:**
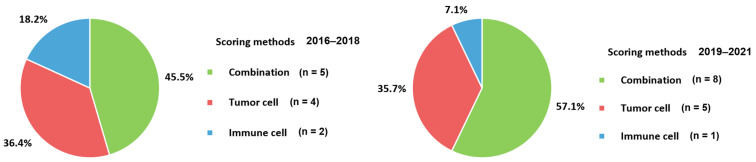
Changes in scoring methods for PD-1 axis. PD-1, programmed cell death protein-1.

**Table 1 cancers-13-05225-t001:** Main characteristics and immunological assays of studies for PD-1, PD-L1, and PD-L2 in metastatic breast cancer.

First Author (Year)	Tumor Type (*n*)	Sample Type	Target	Antibody Clone	Detection Method	Detection System/Assay Producer	Scoring Methods	Cut-Off	Study Country
Adams (2019) [38,39]	TNmBC (170)	Archival and/or freshly collected tumor tissue	PD-L1	22C3	IHC	PD-L1 IHC 22C3 pharmDx (Dako, Agilent, Carpinteria, CA, USA)	Combined positive score	0.01	Multi-national
Adams (2019) [38]	TNmBC (84)	Archival and/or freshly collected tumor tissue	PD-L1	22C3	IHC	PD-L1 IHC 22C3 pharmDx (Dako, Agilent, Carpinteria, CA, USA)	Combined positive score	0.01	Multi-national
Adams (2018) [39]	TNmBC (33)	Archival and/or freshly collected tumor tissue	PD-L1	SP142	IHC	VENTANA PD-L1 (SP142) assay (Ventana Medical System, Roche, Basel, Switzerland)	TC and IC	1% and 1%	USA
Alves (2019) [40]	mBC (41)	FFPE	PD-L1	SP142	IHC	Anti-human PD-L1 rabbit monoclonal (clone SP142, dilution: ready to use; Ventana, Tucson, AZ, USA)	TC	0.01	Portugal
Cimino-Mathew (2016) [41]	mBC (26)	Surgically resected tissue and FFPE	PD-L1	5H1	IHC	In-house	TC	0.05	USA
Dirix (2018) [36]	Any subtype mBC (168)	FFPE (A biopsy or surgical specimen collected within 90 days)	PD-L1	73-10	IHC	PD-L1 IHC 73-10 pharmDx (Dako, Carpinteria, CA, USA)	TC and IC	1% and 10%	Europe and USA
Domchek (2020) [42]	BRCA-mutated mBC (32)	FFPE/serum	PD-L1/soluble PD-L1	SP263/2.7 A4 and 130021	IHC and ECL immunoassay	VENTANA PD-L1 (SP263) assay (Roche, Basel Switzerland)/Anti-PD-L1 capture antibody clone 2.7 A4 (MedImmune) and anti-PD-L1 primary detection antibody clone 130021 (R&D Systems)	TC and IC/concentration of soluble PD-L1	1% and 1%/15.6 pg/mL	Multi-national
Duchnowska (2016) [43]	Brain mBC (84)	FFPE	PD-1/PD-L1/PD-L2	NR	IHC	NBP1-88104 (cytoplasmic) (Novus, Centennial, CO, USA)/AHP2128 (membranous/cytoplasmic) (AbD Serotec, Hercules, CA, USA)/AF1224 (membranous/cytoplasmic) (R&D systems, Minneapolis, MN, USA)	Lymphoid cells/semiquantitative staining H-score/semiquantitative staining H-score	1%/NR/NR	Poland
Emens (2020) [44]	HER2-positive mBC (132)	Pretreatment tumor tissue	PD-L1	SP142	IHC	VENTANA PD-L1 (SP142) assay (Ventana Medical System, Roche, Basel, Switzerland)	IC	0.01	Multi-national
Emens (2019) [45]	TNmBC (115)	Pretreatment tumor tissue	PD-L1	SP142	IHC	VENTANA PD-L1 (SP142) assay (Ventana Medical System, Roche, Basel, Switzerland)	TC and IC	1% and 1%	Europe and USA
Erol (2019) [46]	mBC (47)/mBC (23)	FFPE	PD-L1/PD-L1 methylation	E1L3N	IHC/quantitative real-time PCR	E1L3N monoclonal antibody (Cell Signaling Technology, Danvers, MA, USA)/EZ DNA Methylation-Gold kit (Zymo Research, Irvine, CA USA)	TC	0.05	Turkey
Han (2021) [47]	mBC (208)	FFPE/plasma	PD-L1/soluble PD-L1	ab58810	IHC/ELISA	clone ab58810 (Abcam, Paris, France)/PDCD1LG1 ELISA kit (USCN Life Science, Wuhan, China)	TC/concentration of soluble PD-L1	1%/8.774 ng/mL	China
Loi (2019) [48]	HER2-positive mBC (58)	FFPE	PD-L1	22C3	IHC	QualTek immunohistochemistry assay (QualtTek Molecular Laboratories, Santa Barbara, CA, USA) and Dako IHC 22C3 pharmDx Q² Solutions assay (Q^2^ Solutions, West Lothian, UK)	TC and TCIC	1% and 1%	Multi-national
Manson (2019) [49]	mBC (67)/mBC (83)	FFPE	PD-L1/PD-1	SP263	IHC	anti-PD-L1 rabbit monoclonal antibody (741–4905 (clone sp263, dilution Ventana ready to use; Ventana Medical Systems, Tucson, AZ, USA))/anti-PD-1 mouse monoclonal antibody (ab52587 (NAT105, dilution 1:50, Abcam, Cambridge, UK))	TC and IC/IC	>0%	Netherland
Mitchell (2018) [50]	TNmBC (39)	Tissue	PD-L1	22C3	IHC	PD-L1 IHC 22C3 pharmDx assay (Agilent, Carpinteria, CA, USA)	Combined positive score	0.01	USA
Nanda (2016) [51]	TNmBC (27)	FFPE	PD-L1	22C3	IHC	22C3 antihuman PD-L1 antibody (Merck & Co., Kenilworth, NJ, UK)	TC	0.01	USA
Ogiya (2016) [52]	HER2-positive mBC (14)/TNmBC (11)	FFPE	PD-L1 and PD-L2	ab58810/XX19	IHC	Anti-PD-L1 (polyclonal, ab58810; Abcam, Cambridge, MA, USA)/anti-Pdcd-1L2 (PD-L2, clone XX19; Santa Cruz Biotechnology, Dallas, TX, USA)	Lymphocyte staining	0.1	Japan
Quintela-Fandino (2020) [34]	HER2-negative mBC (24)	FFPE	PD-L1	SP263	IHC	Ventana SP263 assay (Ventana Medical Systems, Inc., Oro Valley, AZ, USA).	NR	0.01	Spain
Rugo (2018) [53]	ER-positive/HER2-negative mBC (25)	FFPE (An archival or newly obtained core or excisional biopsy specimen from a nonirradiated tumor lesion)	PD-L1	22C3	IHC	Prototype assay (QualTek Molecular Laboratories, Goleta, CA, USA) and the 22C3 antibody (Merck & Co. Inc., Kenilworth, NJ, USA)	Tumor proportion score	0.01	Multi-national
Santa-Maria (2018) [54]	ER-positive or TNmBC (18)	Biopsies	PD-L1 and PD-L2	NA	Quantitative real-time PCR	TaqMan probes (Life Technologies, Foster City, CA, USA) and ABI ViiA 7 system (Applied Biosystems, Foster City, CA, USA)	NA	NA	USA
Schmid (2018) [23]	TNmBC (451)	FFPE or fresh pretreatment relapsed-disease tumor tissue	PD-L1	SP142	IHC	VENTANA PD-L1 (SP142) assay (Ventana Medical System, Roche, Basel, Switzerland)	IC	0.01	UK and USA
Schott (2017) [55]	mBC (17)	Circulating epithelial tumor cells	PD-L1/PD-L2	29E.2A3/176611	Immunofluorescence analysis	Anti-human PD-L1 phycoerythrin-conjugated antibody (clone 29E.2A3, BioLegend, San Diego, USA)/anti-human PD-L2 Alexa Fluor^®^ 350 conjugated antibody (clone 176611, novus biologicals, Littleton, CO, USA)	Total number of circulating epithelial tumor cells	NA	Germany
Szekely (2018) [56]	mBC (87)/mBC (27)	Surgically resected tissue and FFPE	PD-L1	E1L3N	IHC	E1L3N XP rabbit monoclonal antibody (Cell Signaling, Danvers, MA, Technology, USA)/Nanostring PanCancer Immune Profiling assay (Nanostring Technologies, Inc., Seattle, WA, USA).	TC	0.01	Europe and USA
Tawfik (2018) [57]	mBC (41)	Core needle biopsy, excision, and FFPE	PD-L1	SP263	IHC	VENTANA PD-L1 (SP263) assay (Roche Ventana Medical Systems, Tucson, AZ, USA)	TC and IC	1% and 10%	USA
Tolaney (2021) [58]	TNmBC (82)	Tissue samples collected at screening	PD-L1	22C3	IHC	PD-L1 IHC 22C3 pharmDx assay (Agilent, Carpinteria, CA, USA)	Combined positive score	0.01	USA
Voorwerk (2019) [59]	TNmBC (66)	FFPE	PD-L1	22C3	IHC	Dako IHC 22C3 pharmDx Q² Solutions assay (Q^2^ Solutions, West Lothian, UK)	TC and IC	NR	Netherland
Yazdanpanah (2021) [60]	mBC (5)	FFPE/serum	PD-L1/soluble PD-L1	E1L3N	IHC/ELISA	E1L3N XP rabbit monoclonal antibody (Cell Signaling Technology, Danvers, MA, USA)/ELISA kit (R&D, Munich, Germany)	TC/concentration of soluble PD-L1	1%/4.52 pg/mL	Iran
Yuan (2019) [35]	mBC (47)	FFPE	PD-L1/PD-1	NR	IHC	NR	TC and stromal cell/IC	1%/1%	China

mBC, metastatic breast cancer; TN, triple-negative; FFPE, formalin-fixed paraffin-embedded; PD-1, programmed cell death protein-1; PD-L1, programmed death ligand-1; PD-L2, programmed death ligand-2; IHC, immunohistochemistry; ECL, electrochemiluminescence; ELISA, enzyme-linked immunosorbent assay; TC, tumor cell; IC, immune cell; NR, not reported.

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
