# Peer review of "Currently Used Laboratory Methodologies for Assays Detecting PD-1, PD-L1, PD-L2 and Soluble PD-L1 in Patients with Metastatic Breast Cancer"

_cancers, 2021, doi:10.3390/cancers13205225_

Round 1
Reviewer 1 Report
The literature review is well done, but some interesting topics are missing. You need emphasize more the substantial differences in the clones and detection methods for the IHC of PD-L1, as the SP142 clone, which must be used only with Ventana, is absolutely not comparable to the others, and the scoring system to be used for first-line immunotherapy on metastatic TNBC is IC (Impassion 130 trial). The separate discussion of this topic (SP142, Ventana and scoring sistem as companion test for Atezolizumab) could make the paper more readable.
On the other hand, Keynote86 trial demonstrated that PD-L1 status was not associated with response to Pembrolizumab; the ORR was 4.8% in PD-L1-positive and 4.7% in PD-L1-negative.
The manuscript does not mention the immunological background, which includes the evaluation of other markers as TILs, CD73, A2A adenosine receptor; the immune microenvironment is predictive also of the response to chemotherapy (Proc Natl Acad Sci USA 2017: 110; 11091-11096. doi; 10.10073/pnas.1222251110 Cancers 2020: 12;2648. doi: 10.3390/cancers 12092648)
Reviewer 2 Report
This review paper is trying to summarize laboratory techniques for detecting PD-1, PD-L1, 535 PD-L2, and soluble PD-L1. The paper reports a few percentages for different conditions (including some figures such as pie-plots). However, the statistical summary/figures lacks sample size (or 95% confidence) information. For example, 10% can be 1/10 or 10/100. There is no statistical justification to make a solid conclusion.
The primary objective (and some 2nd objectives) of this study is not defined clearly. As a result, there is no way to perform a systematic review and meta-analysis.
Reviewer 3 Report
This review proposed by Seri Jeong et al entitled « current Assays for PD-1, PD-L1, PD-L2, and Soluble PD-L1 in Patients with Metastatic Breast Cancer” compare the different assays using immunotherapies in the PD1/2 pathway specifically in the breast cancer, and more precisely in TNBC. In one hand, this paper is not easy to read because of abundance of technical details but in the other hand, these very precise details are, in my mind, the main interest of this review. Indeed, authors did a real hard work to shell the different previous published papers on the field, and propose some attempts to explain the different results (nature of samples (FFPE, fresh…, nature of the antibody used…).
The title is not very clear and should better explain that this review is a comparative assays of results and methodologies.
To aid the readers, I recommend to add some pictures. For example, for explaining the exhaustion. For example, for comparing the structure of PD-L1/L2
table1: numbers after “Adams” are unnecessary but reference number is required
Reviewer 4 Report
Thank you for your valuable work, yet several flaws must be considered.
- The abstract does not include the main aim of the manuscript.
- In table 1, the first authors' names are the same as probably it is the same authors. However, this creates confusion in the readers, so that I suggest inserting reference notes using Mendeley or endnote.
- The manuscript is well-written, yet a personal point of view of the authors is missing. In this light, I suggest the authors to consider the potential clinical implication of PD-1, PD-L1, and PD-L2 in medical practice.

Round 2
Reviewer 2 Report
Thank you for the response. All my comments/concerns have been addressed.
Author Response
We are very appreciated with your constructive comments for our manuscript.
Reviewer 4 Report
Thank you for your valuable work and the implementation you made on the previous version of the manuscript.
My only concern is about some figures, namely figure 5 is cut in the upper part; figure 3 is not presented in good quality, and it appears blurred, as figure 4 and 6; and figure 7 is too small and difficult to read.
